# Reintroduction of resistant frogs facilitates landscape-scale recovery in the presence of a lethal fungal disease

Roland A. Knapp [1,2] ✉, Mark Q. Wilber [3], Maxwell B. Joseph [4,6], Thomas C. Smith [1,2] & Robert L. Grasso [5]

Vast alteration of the biosphere by humans is causing a sixth mass extinction, driven in part by an increase in infectious diseases. The emergence of the lethal fungal pathogen *Batrachochytrium dendrobatidis* (Bd) has devastated global amphibian biodiversity. Given the lack of any broadly applicable methods to reverse these impacts, the future of many amphibians appears grim. The Sierra Nevada yellow-legged frog (*Rana sierrae*) is highly susceptible to Bd infection and most *R. sierrae* populations are extirpated following disease outbreaks. However, some populations persist and eventually recover, and frogs in these recovering populations have increased resistance against infection. Here, we conduct a 15-year reintroduction study and show that frogs collected from recovering populations and reintroduced to vacant habitats can reestablish populations despite the presence of Bd. In addition, the likelihood of establishment is influenced by site, cohort, and frog attributes. Results from viability modeling suggest that many reintroduced populations have a low probability of extinction over 50 years. These results provide a rare example of how reintroduction of resistant individuals can allow the landscape-scale recovery of disease-impacted species, and have broad implications for amphibians and other taxa that are threatened with extinction by novel pathogens.

Human activities are devastating global biodiversity[1], with important implications for ecosystem resilience and human welfare[2]. One consequence of human alteration of the biosphere is an increase in emerging infectious diseases[3,4]. Such diseases pose a severe threat to wildlife populations[5], and have caused dramatic declines and extinctions in a wide range of taxa, including echinoderms, mammals, birds, and amphibians[6–9]. Amphibians are experiencing particularly devastating impacts of disease due to the recent emergence and global spread of the highly virulent amphibian chytrid fungus, *Batrachochytrium dendrobatidis* (Bd)[8,10]. By one estimate, hundreds of species have experienced Bd-caused declines, and numerous susceptible taxa are extinct in the wild[8]. The escalating frequency of disease emergence

and the strength of impacts on wild populations increasingly highlight the critical need for successful mitigation of disease in species conservation.

Following pathogen arrival in a host population, resistance (ability to limit pathogen burden) and tolerance (ability to limit the harm caused by a particular burden) are key mechanisms by which hosts reduce disease impacts[11]. The effectiveness of these mechanisms has direct consequences for the persistence and recovery of host populations in the presence of disease[12]. Host immunity and evolution both play important roles in the development of resistance and tolerance, and are key targets for strategies aimed at mitigating disease impacts in the wild[13].

[1]Sierra Nevada Aquatic Research Laboratory, University of California, Mammoth Lakes, CA 93546, USA. [2]Earth Research Institute, University of California, Santa Barbara, CA 93106-3060, USA. [3]School of Natural Resources, University of Tennessee Institute of Agriculture, Knoxville, TN 37996, USA. [4]Earth Lab, University of Colorado, Boulder, CO 80303, USA. [5]Resources Management and Science Division, Yosemite National Park, El Portal, CA 95318, USA. [6]Present address: Planet, San Francisco, CA 94107, USA. ✉e-mail: roland.knapp@ucsb.edu

Evidence of natural recovery in the many Bd-impacted amphibian populations is surprisingly rare (for notable exceptions, see refs. 14–16). This apparent low resilience to disease may be due to the limited ability of many amphibians to develop Bd resistance or tolerance via either immunity[17–19] or evolution (but see refs. 20,21). In turn, this would also reduce the effectiveness of potential Bd mitigation strategies. For example, a limited ability by amphibians to develop resistance or tolerance suggests that reintroduction of amphibians into sites to reestablish populations extirpated by Bd will often result not in population recovery, but instead in the rapid reinfection and mortality of the introduced animals and/or their progeny[22–25]. If true, the future of many amphibian species threatened by Bd appears bleak.

The Sierra Nevada yellow-legged frog (*Rana sierrae*), is emblematic of the global decline of amphibians caused by Bd[8]. Once the most common amphibian in the high elevation portion of California's Sierra Nevada mountains, USA[26], during the past century this frog has disappeared from more than 90% of its historical range[27]. Due to the severity of its decline and the increasing probability of extinction, *R. sierrae* is now listed as "endangered" under the U.S. Endangered Species Act. The decline of *R. sierrae* was initiated by the introduction of non-native trout into the extensive fishless region of the Sierra Nevada[28,29] starting in the late 1800s. The arrival of Bd in the mid-1900s and its subsequent spread[30] caused additional large-scale population extirpations[31,32]. These Bd-caused declines are fundamentally different from the fish-caused declines because fish eradication is feasible[33] and results in the rapid recovery of frog populations[34,35]. In contrast, Bd appears to persist in habitats even in the absence of amphibian hosts[36], and therefore represents a long-term alteration of invaded ecosystems that amphibians will need to overcome to reestablish populations.

Despite the catastrophic impact of Bd on *R. sierrae*, wherein most Bd-naive populations are extirpated following Bd arrival[31], some populations have persisted after epizootics[37] and are now recovering[16] (Fig. 1). During the epizootic phase, *R. sierrae* are highly susceptible to Bd infection as indicated by very high infection intensities ("load"). In contrast, frogs in recovering populations have reduced susceptibility to Bd infection[16], with loads on adults consistently in the low-to-moderate range[37–39]. This reduced susceptibility is evident even under controlled laboratory conditions[16], indicative of host resistance against Bd infection and not simply an effect of factors external to individual frogs (e.g., environmental conditions). In addition to frogs from recovering populations having higher resistance to Bd infection than those from naive populations, they could also have higher tolerance, but no data are currently available to evaluate this possibility. Therefore, we focus on resistance throughout this paper.

The observed resistance of *R. sierrae* could be the result of several mechanisms that are not mutually exclusive, including natural selection for more resistant genotypes[20,21], acquired immunity[19], and/or inherent between-population differences that pre-date Bd exposure. The possible evolution of resistance and subsequent population recovery is consistent with that expected under "evolutionary rescue", whereby rapid evolutionary change increases the frequency of adaptive alleles and restores positive population growth[40,41]. This intriguing possibility also suggests an opportunity to expand recovery beyond the spatial scale possible under natural recovery by using resistant frogs from recovering populations in reintroductions to vacant habitats[39,42] (Fig. 1).

In the current study, we had three primary objectives. First, determine whether the reintroduction of resistant *R. sierrae*, obtained from populations recovering from Bd-caused declines, allows the reestablishment of populations despite ongoing disease (Fig. 1). We accomplished this using a 15-year large-scale reintroduction study. Second, identify important site, cohort, and individual-level predictors of frog survival using results from the reintroduction study. Third, estimate the probability of persistence for the reintroduced populations over a multi-decadal period, thereby extending our inferences of population recovery beyond the temporal bounds of our reintroduction study (Fig. 1). We did this using a stage-structured matrix model ("population viability model").

Here, we show that the majority of reintroduced populations showed evidence of successful reproduction and recruitment despite the presence of Bd. In addition, post-translocation frog survival is influenced by site, cohort, and frog attributes, but not by Bd load. Finally, results from the population viability model suggest that many reintroduced populations have a low probability of extinction over 50 years. These results indicate that reintroduction of resistant individuals can allow the recovery of species driven to near-extinction by novel infectious diseases.

## Results
### Frog population recovery
To determine whether *R. sierrae* from recovering populations can be used to reestablish extirpated populations, we conducted 24 reintroductions in Yosemite National Park (2006–2020). Each of the reintroductions involved collection of adult frogs from 1 of 3 recovering, Bd-positive "donor" populations and translocating them to 1 of 12 nearby recipient sites (Fig. 2). The 3 donor populations are among the largest *R. sierrae* populations in Yosemite. Following translocation, we estimated adult survival and recruitment of new adults from capture-

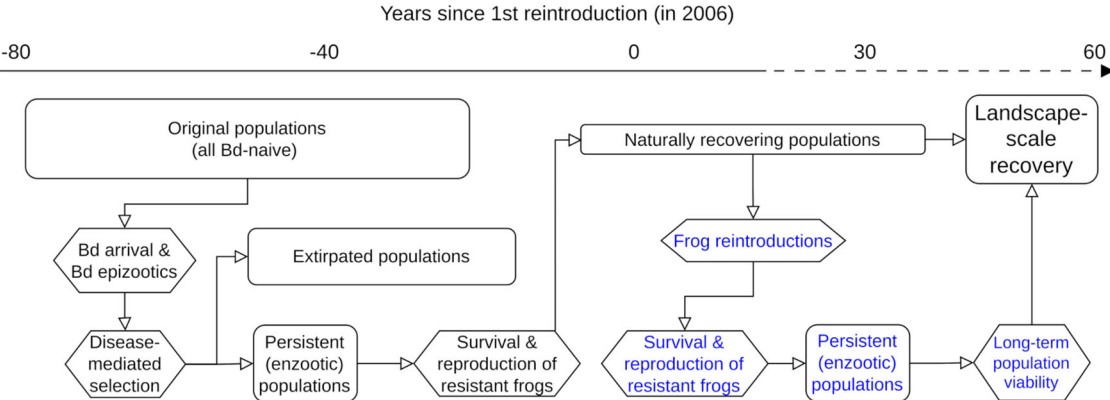

**Fig. 1 | Conceptual model of Bd-caused decline and recovery in *R. sierrae*.** For the recovery portion, the model depicts both natural recovery and facilitated recovery via reintroductions, as well as the linkages between these two pathways. Rectangles and hexagons represent outcomes and processes, respectively. Blue text indicates components that are included in the current study. The general temporal scale of the depicted scenario is provided by the timeline, with the dashed portion indicating a projection into the future.

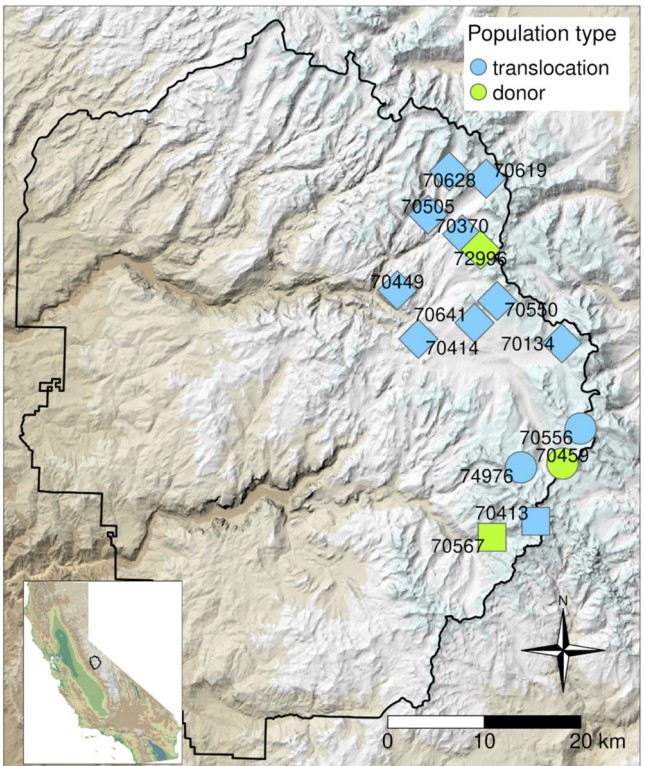

**Fig. 2 | Locations of translocated and donor *R. sierrae* populations.** All populations are in Yosemite National Park (park boundary indicated by black polygon). Symbol shapes indicate the donor population used for each translocation site, and 5-digit numbers identify each donor and translocation site. To obscure the exact locations of populations, random noise was added to all point coordinates. Inset map shows the location of Yosemite within California. In both maps, topography is shown with a grayscale hillshade layer, and elevation is indicated by colors layered over the hillshade. In the absence of hillshading, dark green colors indicate the lowest elevations and white indicates the highest elevation.

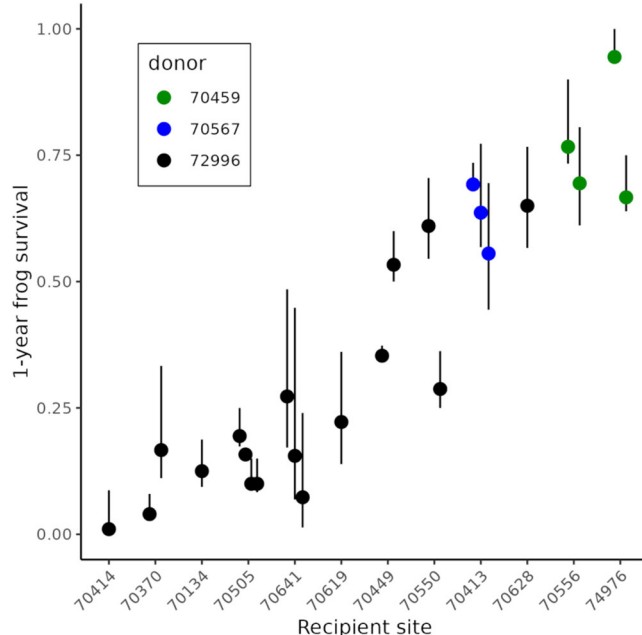

**Fig. 3 | 1-year survival for each cohort of translocated frogs.** Median cohort-level survival was estimated for each of the 12 recipient sites using mrmr CMR models. Error bars show the 95% uncertainty intervals. Sites are arranged along the x-axis using the average of the median 1-year survival per translocation at each site. Point colors indicate the donor population from which frogs in each translocated cohort were collected. When multiple translocations were conducted to a site, points and error bars are slightly offset to avoid overlap.

mark-recapture (CMR) surveys and obtained counts of tadpoles and juveniles from visual encounter surveys (VES). Across all translocation sites, the duration of CMR surveys and VES was 1–16 years (median = 5).

Of the 12 reintroduced populations, 9 (0.75) showed evidence of successful reproduction in subsequent years, as indicated by the presence of tadpoles and/or juveniles. For these 9 populations, one or both life stages were detected in nearly all survey-years following translocation (proportion of survey-years: median = 0.9, range = 0.29–1). These same populations were also those in which recruitment of new adults (i.e., progeny of translocated individuals) was detected. As with early life stages, recruits were detected in the majority of post-translocation survey-years (proportion of survey-years: median = 0.79, range = 0.12–1). In summary, survey results indicate that translocations resulted in the establishment of reproducing *R. sierrae* populations at most recipient sites despite the ongoing presence of Bd.

Bd loads were fairly consistent before versus after translocation, and loads were nearly always well below the level indicative of severe chytridiomycosis (i.e., the disease caused by Bd) and associated frog mortality[31,39] (Supplementary Fig. 1). Although it is possible that the observed relatively small changes in load are a consequence of individuals with high Bd loads dying and therefore being unavailable for sampling during the post-translocation period, the fact that there was little difference in pre- versus post-translocation Bd loads even in those populations that had very high frog survival (70556, 74976 - see below; Supplementary Fig. 1) suggests a true lack of substantial change in Bd load.

The ultimate measure of reintroduction success is the establishment of a self-sustaining population. Given that it can take years or even decades to determine the self-sustainability of a reintroduced population (for an example in *R. sierrae*, see ref. 39), the use of proxies is essential for providing shorter-term insights into reintroduction success and the factors driving it. Results from our CMR surveys allowed us to accurately estimate frog survival, including over the entire CMR time series for each site and during only the 1-year period immediately following translocation. These estimates were made using site-specific models analyzed using the mrmr package[43] (https://snarl1.github.io/mrmr/index.html). We use these estimates to describe general patterns of frog survival in all translocated cohorts, and in an among-site meta-analysis of frog survival to identify important predictors of 1-year frog survival (e.g., Bd load).

Estimates of 1-year frog survival indicate that survival was highly variable between recipient sites, but relatively constant within recipient sites (for the subset of sites that received multiple translocations; Fig. 3). These patterns indicate an important effect of site characteristics on frog survival. In addition, 1-year survival was higher for frogs translocated later in the study period than earlier: 5 of the 7 populations translocated after 2013 had estimated survival ≥ 0.5, compared to only 1 of 5 populations translocated prior to 2013. We suggest this resulted primarily from our improved ability to choose recipient sites with higher habitat quality for *R. sierrae* (see Methods - Frog population recovery - Field methods for details). This increased survival has direct implications for population viability (see Results - Long-term population viability).

The goal of our meta-analysis was to identify important predictors of 1-year frog survival. We were particularly interested in whether Bd load had a negative effect on adult survival, as would be expected if frogs were highly susceptible to Bd infection. This analysis was

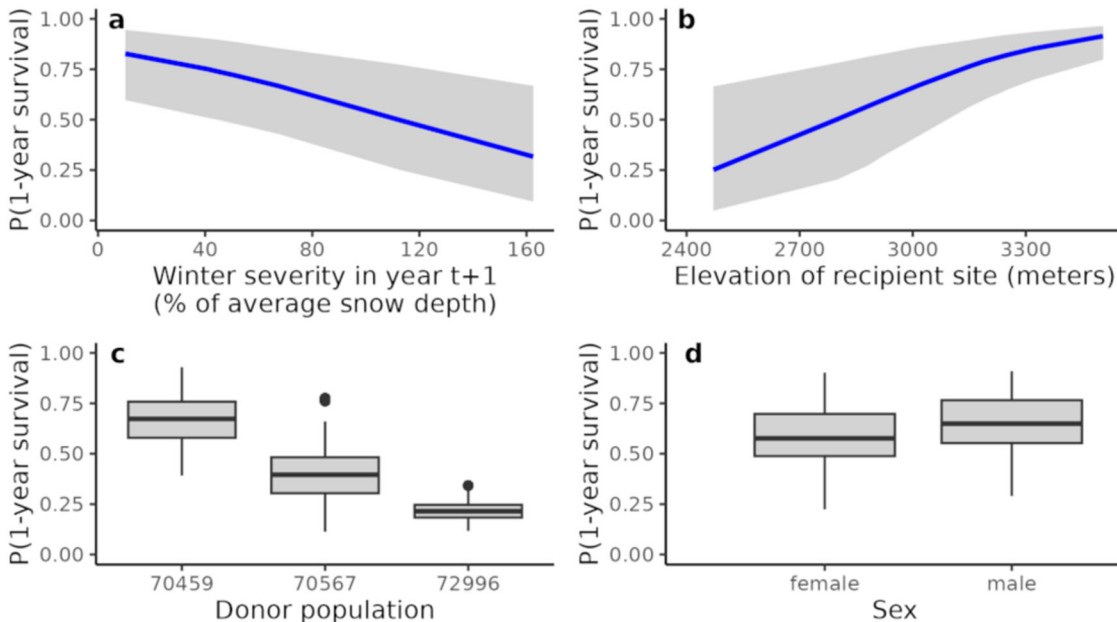

**Fig. 4 | Conditional effects of predictors of 1-year frog survival.** Results are from the among-site meta-analysis and 1-year frog survival is expressed as a probability. (**a**) winter severity in the year following translocation, (**b**) elevation of recipient site, (**c**) donor population, and (**d**) sex. In (**a**, **b**), blue lines are medians and gray ribbons are 95% uncertainty intervals for the posterior distributions of the predicted effects. In (**c**, **d**), box plots show medians (horizontal line), first and third quartiles (hinges), largest and smallest values within 1.5x interquartile range (whiskers), and values outside the 1.5x interquartile range (dots) for the posterior distributions of the predicted effects.

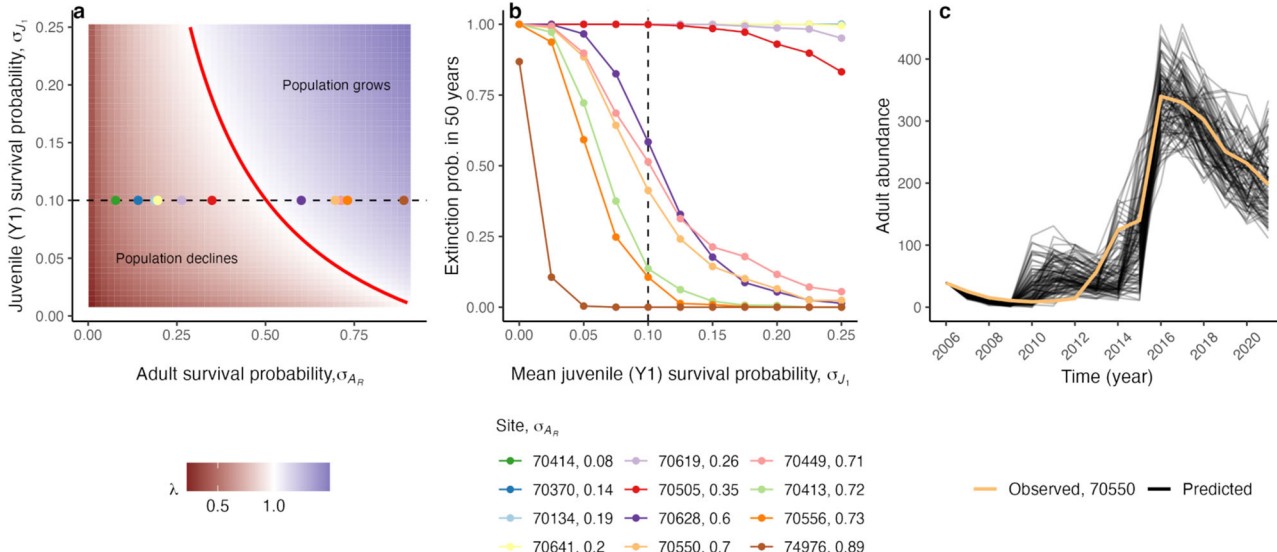

**Fig. 5 | Estimated growth rates and extinction probabilities of reintroduced populations. a** Predicted long-run growth rate $\lambda$ for different values of yearly adult survival probability $\sigma_{A_R}$ and year-1 juvenile survival probability $\sigma_{J_1}$, given the para-meterized, deterministic model. Colored points show the predicted $\lambda$ values for the twelve translocated populations when year-1 juvenile survival probability is $\sigma_{J_1} = 0.10$ (indicated by the dashed line). The red line shows where $\lambda = 1$. Note that the point for 70413 is mostly hidden behind other points. **b** Predicted 50-year extinction probabilities of the 12 reintroduced populations, given demographic stochasticity, environmental variability in $\sigma_{J_1}$, and different mean values of $\sigma_{J_1}$. There are 6 lines at extinction probability = 1, 5 of which (70414–70619) are partially or completely hidden beneath the line for 70505. **c** 100 simulated trajectories (black lines) from the population viability model that most closely matched the observed abundance trajectory of adult amphibians at site 70550 (light orange).

conducted in a Bayesian framework and included a diversity of site, cohort, and individual-level characteristics as predictors and 1-year frog survival (Fig. 3) as the response variable. The best model of 1-year frog survival identified several important predictors, but Bd load at the time of translocation was not among them (Supplementary Fig. 2). Instead, important predictors included winter severity in the year following translocation (snow_t1), site elevation, and donor population (Fig. 4, Supplementary Fig. 2). Males had somewhat higher survival than females, but this effect was small (Fig. 4, Supplementary Fig. 2). The absence of Bd load as an important predictor of frog survival is consistent with frogs in recovering populations having sufficient resistance to suppress Bd loads below harmful levels.

In summary, results from our frog translocation study indicate that (i) translocations produced relatively high 1-year survival of translocated adults, as well as reproduction and recruitment, at the majority of recipient sites, (ii) 1-year survival of adults is influenced by site characteristics, weather conditions, and donor population (but not Bd load), and (iii) based on the relatively small changes in Bd load after translocation, loads appear more strongly influenced by frog characteristics (e.g., resistance) than site characteristics. Together, these results indicate that frogs translocated from recovering populations can maintain the benefits of resistance in non-natal habitats. In addition, in 3 locations where longer CMR time series allowed us to assess the survival of new adults recruited to the population, naturally-recruited adults had equivalent or higher survival probabilities than the originally translocated adults (Supplementary Fig. 3). This suggests that frog resistance is maintained across generations. All of the conditions described above are supportive of population establishment and long-term population growth.

### Long-term population viability

Results from the frog translocation study are suggestive of population establishment. However, a decade or more of surveys may be necessary to confirm that populations are in fact self-sustaining[39]. To extend our inferences of population establishment beyond those possible from the site-specific CMR data, we developed a population viability model. Specifically, to test whether the observed yearly adult survival probabilities in translocated populations were sufficient for long-term viability, we built a stage-structured matrix model that captured known frog demography and included demographic and environmental stochasticity. We parameterized the model using CMR data from translocated populations and known life history values in this system (Supplementary Table 1). Results from a sensitivity analysis are provided in Supplementary Fig. 4.

Given observed yearly adult survival probabilities of translocated frogs (from site-specific mrmr CMR models; provided in legend of Fig. 5b) and a yearly survival probability of the year-1 juvenile class ($\sigma_{J_1}$) greater than 0.10, at least six of twelve translocated populations are likely to experience a long-run growth rate $\lambda$ greater than 1 in the presence of Bd (Fig. 5a; median predicted $\lambda$ ranges from 1.07 to 1.28 for these six populations). These six populations all had observed yearly adult survival greater than 0.5. As year-1 juvenile survival probability increased above 0.2, the deterministic long-run growth rate of seven of twelve populations was greater than 1 (Fig. 5a).

Even when incorporating (i) demographic stochasticity and (ii) environmental stochasticity in year-1 juvenile survival (the transition that we expect to be the most subject to environmental variability in the presence of Bd), populations with high adult survival are likely to persist over a 50 year time horizon. Our model predicted that, following a single introduction of 40 adult individuals into a population, the six populations with the highest adult survival probabilities ($\sigma_{A_R} > 0.5$) had 50-year extinction probabilities of less than 0.5 when the average year-1 juvenile survival was greater than 0.10 (Fig. 5b). This indicates strong potential for long-term persistence in the presence of Bd and environmental variability in juvenile survival. In contrast, for the six populations where yearly adult survival probability $\sigma_{A_R} < 0.5$, extinction probability over 50 years was always predicted to be > 50% regardless of the value of mean year-1 juvenile survival between 0 and 0.25. To test the validity of our model predictions, we demonstrated that our stochastic model could describe the general recovery trajectory of our translocated population with the longest survey history (Fig. 5c; population 70550, surveyed for 16 years).

In summary, our model demonstrates that given observed yearly adult survival probabilities of translocated frogs, 50% of our translocated populations have a high probability of population growth and long-term viability in the presence of Bd. This is likely a conservative estimate because there is evidence that naturally-recruited adults have higher survival probability than translocated adults (Supplementary Fig. 3), but we considered these probabilities to be equal in all but three of our populations where we had sufficient data to distinguish these different probabilities.

## Discussion

Disease-induced population declines are decimating global biodiversity[5], but broadly-applicable strategies to recover affected species are generally lacking (e.g.,[13]). Here, we tested the possibility that populations of resistant individuals from naturally recovering populations can be used to reestablish extirpated populations of the endangered *R. sierrae* in the presence of a highly virulent pathogen (Bd). Our results indicate (i) the capacity of reintroduced populations to become established and eventually recover despite ongoing disease, (ii) that post-translocation frog survival is influenced by site, cohort, and individual-level characteristics, and (iii) that 50% of the reintroduced populations are likely to persist over a 50-year period. In light of the generally low success rate of amphibian reintroduction efforts[44], our success in reestablishing *R. sierrae* populations via reintroduction of resistant individuals is striking, and even more so given that *R. sierrae* was driven to near-extinction by Bd. Collectively, these results provide a rare example of amphibian recovery in the presence of Bd, and have important implications for the conservation and recovery of amphibians and other taxa worldwide that are endangered by escalating impacts from emerging infectious diseases.

Previous field studies in *R. sierrae* show that frog-Bd dynamics and frog survival in the presence of Bd are fundamentally different between naive and recovering populations. Following the arrival and establishment of Bd in previously-naive populations, adult frogs develop high Bd loads that lead to mass die-offs[31]. In contrast, in recovering populations adult frogs typically have low-to-moderate and relatively constant Bd loads and mass die-offs are not observed[37,38] (see also Supplementary Fig. 1). The differences in Bd load of frogs from naive and recovering populations are also observed in controlled laboratory studies (see Figure 4 in[16]), and clearly indicate that frogs from recovering populations exhibit resistance against Bd infection. Finally, reintroductions of *R. sierrae* collected from Bd-naive populations and reintroduced into Bd-positive sites have consistently failed due to the development of high Bd loads and resulting high frog mortality, suggesting the importance of frog resistance in population reestablishment in the presence of Bd.

In the current study, reintroduction of resistant *R. sierrae* was remarkably successful in reestablishing populations in the presence of Bd. Of the 12 translocated populations, approximately 80% showed evidence of both successful reproduction and recruitment of new adults. Year-1 survival for 12 of the 24 translocated cohorts exceeded 50%, and > 70% of translocated cohorts had survival above this 50% level when the earliest translocations are excluded (i.e., translocations conducted when methods were still being refined; see Methods - Frog population recovery - Field methods for a brief description of these refinements). The fact that the relatively low Bd loads and correspondingly high frog survival were both maintained when frogs were moved from donor populations to recipient sites indicates that these qualities of naturally-recovering populations were not solely an effect of site characteristics, but were also strongly influenced by intrinsic characteristics of frogs, including resistance. Although it could be argued that the relatively invariant Bd loads before versus after translocation are a consequence of similar pathogen pressure in the donor and reintroduced populations, this is at odds with the fact that in the first year after translocation frog densities are typically 1–2 orders of magnitude lower in the reintroduced versus donor populations and pathogen pressure should follow a similar pattern. In addition to the maintenance of Bd load and frog survival between natal and recipient sites (Supplementary Fig. 3), the relatively high survival of

translocated frogs was maintained in their progeny, as expected if resistance has a genetic basis.

Results from the population viability model were also encouraging. In particular, translocated populations with > 50% survival in the first year post-translocation were predicted to have a low probability of extinction over 50 years (probability of extinction < 0.5 when year-1 juvenile survival probability was greater than 0.10). The viability model highlighted the important role of frog survival in affecting long-term population viability, and allowed us to extend the temporal scale of our study beyond the years covered by our post-translocation surveys. These long-term forecasts are important, given that reintroduced *R. sierrae* populations may often take decades to achieve our ultimate goal of self-sustainability[39]. Making well-supported projections about the long-term outcome of reintroduction efforts from shorter-term information is crucial for adaptive management of species reintroduction programs[45], including the one we are carrying out for *R. sierrae*. Specifically, the combined results from our reintroduction study and viability modeling suggest that survival of frogs in the first year following translocation is an effective proxy of longer-term survival and population viability. In addition, given the repeatability of frog survival at a site, 1-year frog survival also serves as an effective proxy of site quality (i.e., the ability of a site to support high frog survival and a viable frog population over the long term). This proxy of site quality is important in the *R. sierrae* system because accurately predicting the ability of a site to support a viable frog population a priori remains difficult, even after conducting 24 translocations over 16 years.

The results of the meta-analysis indicate the influence of several variables on adult survival, and by extension on population viability. Although we assume that the effects of winter severity and frog sex are direct effects of the variables themselves, the elevation and donor population variables are likely serving as proxies for a range of factors. For example, elevation is correlated with several attributes that may influence frog survival, including the presence of boulder habitat (positive correlation) that is important for frog overwintering, abundance of frog predators including garter snakes (negative correlation), and frog developmental rates (negative correlation). The latter can influence the duration of frog life stages that have differential susceptibility to chytridiomycosis (see below). The duration of the tadpole and juvenile stages increases with elevation, but the fact that long-run population growth rate from our viability model is relatively insensitive to variation in these durations suggests that developmental rates of early frog life stages have minimal effect on the predicted long-term viability of reintroduced populations. The mechanisms underlying the effect of donor population on the probability of post-translocation frog survival is intriguing, and could indicate population-specific fitness differences of frogs in the presence of Bd. Although our study design did not allow us to completely separate the effects of donor population and recipient site on 1-year adult survival, we are currently doing so using a modified study design. Specifically, we are conducting reintroductions in which frogs from several donor populations are added concurrently to a single site. Survival of frogs from each donor population is estimated using CMR methods and the contribution of each donor population to subsequent generations is quantified using genetic methods.

Despite the demonstrated resistance of adult *R. sierrae* against Bd infection, individual and population-level impacts of Bd are still evident. In an earlier study of 2 of our 12 translocated populations[39], Bd infection and load had detectable effects on the survival of adults and may have influenced population establishment (sites referred to as "Alpine" and "Subalpine" in[39] are identified as "70550" and "70505" in the current study). Applying similar analyses to all 12 of our translocated populations would likely provide a broader perspective of the ongoing effect of Bd. In addition to these important but relatively subtle effects of Bd on adults, the impacts on younger life stages are more apparent. *R. sierrae* immediately following metamorphosis

("metamorphs") are highly susceptible to Bd infection[46] and as a result experience high mortality[32]. This high susceptibility of metamorphs is documented in numerous species of anurans, and may result from the poorly developed immune system characteristic of this life stage[47]. In both naturally recovering and reintroduced *R. sierrae* populations, we suggest that the high mortality of metamorphs is an important limitation on subsequent recruitment of new adults. Therefore, although adult *R. sierrae* appear relatively resistant, Bd infection continues to have important limiting effects on recovering populations (see also ref. [48]). The ongoing high susceptibility of metamorphs to chytridiomycosis also highlights the critical need to develop better methods of monitoring this vulnerable life stage, the results of which could significantly advance the conservation and recovery of Bd-impacted amphibians.

Our successful use of translocations to reestablish *R. sierrae* populations despite ongoing Bd infection is particularly important because natural recovery alone appears insufficient to allow reestablishment of populations at the landscape scale. Naturally recovering *R. sierrae* populations in Yosemite are relatively rare and are often isolated from other such populations and from suitable but vacant habitats. This isolation is the consequence of both natural fragmentation due to rugged geography and the ongoing presence of introduced, predatory trout in the majority of lakes and streams[49]. Collectively, these factors often prevent naturally recovering *R. sierrae* populations from expanding via dispersal of frogs to nearby unoccupied habitats. Translocations allowed us to overcome these dispersal limitations and reestablish populations in sites that would otherwise remain unoccupied for the foreseeable future. Of particular note are the translocations that we conducted as part of the current study into areas from which *R. sierrae* were completely extirpated, but instead of single isolated habitats these areas contained habitats composed of interconnected fishless lakes, ponds, marshes, and/or streams. Successful reestablishment of one *R. sierrae* population in such areas has produced a source of frogs from which adults and juveniles are now dispersing into adjacent habitats. This process of translocation, establishment, and subsequent dispersal has significant potential to eventually allow the reestablishment of robust metapopulations that may be more resilient to current and future natural and anthropogenic impacts than more fragmented populations (e.g., ref. [50]).

The promising example provided by our study of successful reestablishment of *R. sierrae* despite ongoing Bd infection may provide an important roadmap for ongoing and future efforts to recover the hundreds of Bd-impacted amphibian species globally. Importantly, in addition to the natural recovery documented for *R. sierrae*[16], other amphibian species are also showing evidence of post-epizootic recovery in the presence of Bd[14,15] and suggest the possibility of also using animals from these recovering populations to reestablish extirpated populations. As with *R. sierrae*, the feasibility and long-term success of such efforts will depend on the availability of robust donor populations containing individuals that have sufficiently high resistance to allow frog survival and population growth in the presence of Bd. We suggest that several components of recovery efforts could play important roles in maximizing their success. First, an adaptive management approach such as the one we utilized, in which the outcomes of recovery actions are rigorously assessed and this information is used to reduce uncertainties and inform future actions, will often be essential[51]. Second, given the apparent rarity in amphibians of evolution of resistance against Bd infection, developing an improved understanding of the factors facilitating or limiting the development of frog resistance and the effectiveness of selective breeding programs and other genetic interventions to enhance frog resistance[52] may be particularly important. Finally, although we did not quantify the genetic structure of the donor and translocated populations of *R. sierrae* as part of the current study, conducting genetic monitoring may be important to identify sub-optimal genetic outcomes during

establishment of translocated populations (e.g., genetic bottlenecks) and allow timely implementation of actions to mitigate such effects[53].

In conclusion, we now have a proven strategy to reestablish extirpated *R. sierrae* populations. However, recovery across their large historical range will require substantial resources over many decades. The results of this study provide a hopeful starting point for that endeavor and other future efforts worldwide.

## Methods

### Frog population recovery

**Field methods.** The collection of data from live *R. sierrae* described here complies with all ethical regulations and was approved under the University of California-Santa Barbara Institutional Animal Care and Use Committee (IACUC) protocol number 478. For the 24 translocations we conducted, we identified donor populations from which adult frogs (≥ 40 mm snout-vent length) could be collected based on several years of VES and skin swab collections[16], and results from population genetic analyses[54]. The populations that we selected contained hundreds of *R. sierrae* adults and thousands of tadpoles. These relatively high abundances were the result of recent increases following previous Bd-caused declines[16]. As is typical for recovering *R. sierrae* populations, Bd prevalence in the donor populations was high (0.69–0.96) and Bd load (median $\log_{10}$(load) = 3.06–3.78 ITS copies) was two or more orders of magnitude below the level at which frog mortality is expected[31,39] ($\log_{10}$(load) ≈ 5.78 ITS copies). Recipient sites to which frogs were translocated were chosen based on previous *R. sierrae* presence (determined from VES and/or museum records) or characteristics that suggested high quality habitat for this species[55]. At the beginning of this study, we had a relatively limited understanding of the factors that affect habitat quality. In subsequent years, we improved our site selection process by incorporating new information about important habitat features, in particular, overwintering habitats such as submerged boulders and overhanging banks. *R. sierrae* were absent from all recipient sites prior to the first translocation. This determination was based on VES conducted at recipient sites over several years (see additional details below regarding frog detectability during VES).

We conducted 1–4 translocations per site (Figs. 2, 3) and each translocated cohort included 18 to 99 frogs (median = 30). In preparation for each translocation, adult frogs were collected from the donor population and measured, weighed, swabbed, and PIT tagged. Frogs were transported to the recipient site either on foot or via helicopter. Following release, we visited translocated populations approximately once per month during the summer active season to conduct diurnal CMR surveys and VES (summer active season is generally July-August but can start as early as May and end as late as September; range of survey dates = May-25 to Sep-29, range of translocation dates = Jun-28 to Sep-02; median number of visits (i.e., primary periods; see below) per summer = 2, range = 1–10). CMR surveys allowed estimation of adult survival, recruitment of new adults, and adult population size. VES provided estimates of tadpole and juvenile abundance. During 2006-2012, we conducted CMR surveys on a single day per site visit (primary period), during which we searched all habitats repeatedly for adult frogs. Frogs were captured using handheld nets, identified via their PIT tag (or tagged if they were untagged), measured, weighed, swabbed, and released at the capture location. During 2013-2022, we generally used a robust design in which all habitats were searched during several consecutive days (median number of secondary periods per primary period = 3; range = 3–7), and frogs were processed as described above. However, when the number of frogs detected on the first survey day was zero or near zero, we typically conducted only a single-day CMR survey. When using a robust design, within a primary period, frogs that were captured during more than one secondary period were measured, weighed, and swabbed

during the first capture, and during subsequent captures were only identified and released.

During each site visit, we conducted VES either immediately before CMR surveys or during the first day of CMR surveys. VES was conducted by walking the entire water body perimeter, first 100 m of each inlet and outlet stream, and any fringing ponds and wetlands, and counting all *R. sierrae* tadpoles and juveniles. These *R. sierrae* life stages have high detectability, and counts are highly repeatable and provide estimates of relative abundance[29].

**Frog counts and reproductive success.** For each of the translocated populations, we used the presence of tadpoles and/or juveniles from VES and counts of new recruits (i.e., untagged adults) in CMR surveys to provide two measures of successful reproduction. To calculate the proportion of years in which tadpoles/juveniles were present at a site, we excluded surveys conducted in the year of the initial translocation to that site. This exclusion accounted for the fact that all translocations were conducted after the breeding period and reproduction would therefore not occur until the following year. Similarly, to calculate the proportion of years in which new recruits were present at a site, we excluded surveys conducted during the 3 years following the initial translocation. This accounted for the multi-year tadpole and juvenile stages in *R. sierrae* (Supplementary Table 1).

**Estimation of frog survival and abundance.** For each translocation site, we estimated survival of translocated frogs, recruitment of new frogs into the adult population, and adult population size using a site-specific Bayesian open-population Jolly-Seber CMR model with known additions to the population (i.e., translocated cohorts), as implemented by the mrmr package[43] and using R Statistical Software[56] (v4.4.4) (see Supplementary Methods - Frog population recovery - CMR model structure for details). Briefly, the model tracks the states of $M$ individuals that comprise a superpopulation made up of real and pseudo-individuals[39]. The possible states of individuals include "not recruited", "alive", and "dead". The possible observations of individuals include "detected" and "not detected". We assume that individuals that are in the "not recruited" or "dead" states are never detected (i.e., there are no mistakes in the individual PIT tag records). We also assume that new recruits were the result of within-site reproduction and not immigration from adjacent populations. This assumption is justified by the fact that no *R. sierrae* populations were present within several kilometers of the translocation sites. For all models, we used mrmr defaults for priors, number of chains (4), and warmup and post-warmup iterations (2000 for each). We evaluated convergence of the Markov chain Monte Carlo (MCMC) algorithm using trace plots and Gelman-Rubin statistics (Rhat).

**Predictors of post-translocation frog survival.** To identify important predictors of frog survival following translocation, we used multilevel Bayesian models[57,58]. Included predictor variables describe characteristics of sites, translocated cohorts, and individuals (Bd load, sex, frog size, site elevation, winter severity in the year of translocation, winter severity in the year following translocation, donor population, day of year on which a translocation was conducted, and translocation order). We used 1-year post-translocation survival estimates from CMR models as the response. Estimated survival was rounded to integer values to produce a binary outcome, and modeled with a Bernoulli distribution. Group-level (random) effects included site_id, translocation_id, or translocation_id nested within site_id. We performed all analyses with the rstanarm package[59] and R Statistical Software[56] (v4.4.4). For all models, we used default, weakly informative priors, four chains, and 5000 iterations each for warmup and post-warmup. We checked MCMC convergence using trace plots and Rhat, and evaluated model fit using leave-one-out cross-validation[60], as

implemented by the loo package[61]. (See Supplementary Methods - Frog population recovery - Among-site survival modeling for details.)

**Changes in Bd load following translocation.** We analyzed skin swabs using standard Bd DNA extraction and qPCR methods[62] (see Supplementary Information - Frog population recovery - Laboratory methods for details). To assess the magnitude of changes in Bd load on frogs following translocation, we compared Bd loads measured before versus after translocation. Before-translocation loads were quantified using skin swabs collected from all to-be-translocated frogs at the donor site on the day before or the day of the translocation. After-translocation Bd loads were based on all swabs collected from translocated frogs at the recipient site in the year of and the year following translocation. Individual frogs and their associated Bd loads were included in the dataset only if frogs were captured at the recipient site at least once during the 1-year period following translocation.

## Population viability modeling

**Model description.** To determine the implications of observed 1-year adult survival on the long-term viability of populations established via translocation, we developed a population model for *R. sierrae*. Our central question was: How does the magnitude and variation in observed adult survival probability across translocated populations affect the long-term persistence probability of populations? We developed a model that tracked seven state variables of a frog population: abundance of translocated adults ($A_T$), abundance of adults naturally recruited into the population ($A_R$), abundance of first-year tadpoles ($L_1$), abundance of second-year tadpoles ($L_2$), abundance of third-year tadpoles ($L_3$), abundance of first-year juveniles ($J_1$), and abundance of second-year juveniles ($J_2$). We divided adults into two classes $A_T$ and $A_R$ because there is evidence that the survival probability of translocated adults and naturally recruited adults differs (Supplementary Fig. 3). The modeled population included both male and female frogs.

We modeled the dynamics of these seven state variables using a discrete-time, stage-structured model where a time step is one year. The dynamics are given by

decision because (i) translocated populations are infected with Bd at high prevalence[39], and (ii) host density does not seem to play a significant role in multi-year Bd infection dynamics in this system[63]. Thus, ignoring Bd infection dynamics and instead assuming all host vital rates are in the presence of high Bd prevalence significantly simplifies the model without much loss of realism. Additional details are provided in Supplementary Methods - Population viability modeling - Incorporating yearly variability in survival rates and Estimating model parameters.

**Model analysis.** After parameterizing our model with CMR-estimated adult frog survival probabilities and other known vital rates (Supplementary Table 1), we performed four analyses. First, we compute the long-run growth rate $\lambda$ for each of our 12 translocated populations to determine if the populations were deterministically predicted to grow or decline in the long-run. Second, we compute the elasticity and sensitivity of $\lambda$ to nine model parameters to quantify how much changes in these parameters affected the long-run growth rate (Supplementary Fig. 4). This also helped us determine where in the model environmental variation in juvenile survival would have the largest effects on population dynamics. Third, we included demographic stochasticity and environmental stochasticity in $\sigma_{J_1}$ in our model and simulated the 50-year viability (i.e., 1 - extinction probability) of populations given an introduction of 40 adult individuals into an unoccupied habitat. Finally, we fit our model to our longest translocation trajectory to confirm that our model could reasonably reproduce the observed recovery trajectories of *R. sierrae* following reintroductions. Additional details are provided in Supplementary Methods - Population viability modeling - Model analysis and simulation.

## Reporting summary
Further information on research design is available in the Nature Portfolio Reporting Summary linked to this article.

## Data availability
The data used in this study are available at https://doi.org/10.5281/zenodo.13851590[64]. This includes the estimated survival of each rein-

$$
\begin{bmatrix} L_1 \\ L_2 \\ L_3 \\ J_1 \\ J_2 \\ A_R \\ A_T \end{bmatrix}(t+1) = \begin{bmatrix} 0 & 0 & 0 & 0 & 0 & \sigma_{A_R}p_F\rho F & \sigma_{A_T}p_F\rho F \\ \sigma_{L1}p_{L1} & 0 & 0 & 0 & 0 & 0 & 0 \\ 0 & \sigma_{L2}p_{L2} & 0 & 0 & 0 & 0 & 0 \\ \sigma_{L1}(1-p_{L1}) & \sigma_{L2}(1-p_{L2}) & \sigma_{L3} & 0 & 0 & 0 & 0 \\ 0 & 0 & 0 & \sigma_{J1}p_{J1} & 0 & 0 & 0 \\ 0 & 0 & 0 & \sigma_{J1}(1-p_{J1}) & \sigma_{J2} & \sigma_{A_R} & 0 \\ 0 & 0 & 0 & 0 & 0 & 0 & \sigma_{A_T} \end{bmatrix} \begin{bmatrix} L_1 \\ L_2 \\ L_3 \\ J_1 \\ J_2 \\ A_R \\ A_T \end{bmatrix}(t) \tag{1}
$$

The parameters in this model are yearly survival probability $\sigma_.$ (the subscript "·" indicates a particular state variable), probability that a female frog reproduces in a given year $p_F$, the percentage of adults that are female $\rho$, number of eggs produced by a female frog in a year that successfully hatch $F$, probability of a first-year tadpole remaining as a tadpole $p_{L1}$, probability of a second-year tadpole remaining as a tadpole $p_{L2}$, and probability of a first-year juvenile remaining as a juvenile $p_{J1}$. First-year juvenile survival $\sigma_{J1}$ is the parameter that we think is most influenced by environmental stochasticity.

In this model we ignore density-dependent recruitment because we were interested in the growth of the population from an initial reintroduction and whether this growth was sufficient to prevent extinction over 50 years following the introduction. We also did not directly consider the dynamics of Bd in this model. We made this

troduced frog cohort, data used to identify predictors of 1-year frog survival, data used to estimate growth rates and extinction probabilities of reintroduced populations, Bd loads on frogs before and after translocation, and survival probabilities of translocated versus naturally recruited adults. For the relevant figures, source data are provided in a Source Data file. The CMR datasets from each reintroduction site that we used to calculate post-translocation frog survival are available at https://doi.org/10.5281/zenodo.13851652[65]. Source data are provided with this paper.

## Code availability
R code to replicate our analyses is available at https://doi.org/10.5281/zenodo.13851590[64]. R code to analyze the CMR datasets is available at https://doi.org/10.5281/zenodo.13851652[65].

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

## Acknowledgements

We thank the following for important contributions to this study: E. Hegeman and A. Lindauer (project and data management, field and laboratory work); A. Barbella and K. Rose (laboratory work); numerous summer technicians (field work); staff at Sequoia-Kings Canyon and Yosemite National Parks, Inyo and Sierra National Forests, California Department of Fish and Wildlife, U.S. Fish and Wildlife Service, Sierra Nevada Aquatic Research Laboratory (DOI: 10.21973/N3966F), and University of California-Santa Barbara Institutional Animal Care and Use Committee (research permits, logistical support, and/or field assistance). This project was supported by grants from the National Park Service (to R.A.K.), Yosemite Conservancy (to R.A.K.), and National Science Foundation (EF-0723563, to C. Briggs; DEB-1557190, to C. Briggs; DEB-2133401, to M.Q.W; and DBI-2120084, to C. Richards-Zawacki).

## Author contributions

R.A.K., M.Q.W., M.B.J., T.C.S., and R.L.G designed the research. R.A.K., M.B.J., T.C.S., and R.L.G. contributed to data collection; M.B.J. contributed new analytic tools; R.A.K. and M.Q.W. conducted the analyses. R.A.K. and M.Q.W. wrote the manuscript with input from M.B.J., T.C.S., and R.L.G.

## Competing interests

The authors declare no competing interests.
