## [Transparent Peer Review file · Nature Communications]

Reintroduction of resistant frogs facilitates landscape-scale recovery in the presence of a lethal fungal disease

Corresponding Author: Dr Roland Knapp

Version 0:

Reviewer comments:

Reviewer #1

(Remarks to the Author)

This is a brilliant manuscript that advances our understanding and ability to mitigate the impact of the worst disease to impact vertebrates, chytridiomycosis. The main findings are that reintroduced frogs sourced from populations of the Sierra Nevada yellow-legged frog (*Rana sierrae*) that are recovering from initial outbreaks and have increased resistance against chytridiomycosis are able to reestablish populations in the presence of the pathogen, Bd. The authors also investigated factors affecting reintroduction success and found that site, cohort, and frog attributes were important. They modelled the viability of reintroduced populations suggesting that many have a low probability of extinction over 50 years.

The work has major significance to the growing fields of chytridiomycosis, wildlife diseases, mitigation of threatening processes, conservation, reintroduction of species and related fields. The work is original and the first to demonstrate reintroduction of resistant frogs as a viable long term solution for recovering amphibian populations from the impacts of chytridiomycosis. The evidence presented supports the conclusions of the authors and the methodology is sound. The figures are excellent.

Some suggestions for improvement of the manuscript:

I appreciate the arguments made in favour of frog resistance rather than site characteristics being the reason for translocation success. Perhaps the strongest argument is that non resistant frogs were extirpated from these sites previously and would be expected to be extirpated again unless they had evolved resistance and this could be stated in the discussion.

The manuscript could possibly be improved by recognising that although many species have failed to evolve resistance, this may occur over a longer time frame or alternatively, species could be assisted to evolve resistance through selective breeding programs etc (Kosch et al 2022).

Similarly, the manuscript could perhaps expand on the barriers to natural recovery and spread of resistant populations and the importance of considering elevational disease gradients and connectivity when choosing reintroduction sites (Bell et al 2020).

Perhaps the authors could make a point in the discussion regarding the importance of sustained effort and using adaptive management techniques. This approach ensured they improved their ability to choose recipient sites with higher habitat quality and ultimately led to improved translocation success.

Finally, perhaps the conclusions could be a little more hopeful given the study's results along with other current efforts at mitigating the threat of chytridiomycosis (Berger et al 2024, Waddle et al 2024).

Congratulations to the authors on a very impressive body of work and most importantly, wonderful results for amphibian conservation. Extremely well done!

All the best
Lee Skerratt

(Remarks to the Author)

This is an extremely important paper – the authors have conducted extraordinary conservation work on an amphibian species known to be highly susceptible to the lethal fungal pathogen (*Batrachochytrium dendrobatidis*; “Bd”). Their efforts to reintroduce the Sierra Nevada yellow-legged frog (*Rana sierrae*) to locations where it had formally been extirpated using adults from naturally recovering populations is extremely promising. To my knowledge, this paper constitutes the largest reintroduction effort to date for an amphibian challenged by Bd, both in terms of the number of reintroductions and their subsequent success. I have few minor comments (below) and highly recommend publishing this paper. It will certainly be highly cited and serve as a model for other amphibian systems impacted by the pathogen. Additionally, I think the methods and associated data analyses are well explained and appropriate (see minor comments/suggestions below).

Minor comments/suggestions: (these are in no order)

1. Line 90-98. It seems like the sentences that discuss the three primary objectives may not be complete sentences? It seems like they might need another word: e.g., First, ‘we’ determined whether... Second, ‘we’ identified.... Etc. Maybe authors mean to provide a list. We had three primary objectives (1) determine whether...; (2) identify important site,... (3) estimate the probability of persistence...but that is a lot to include in a single sentence, so I understand (and like) breaking it apart, but I am just not sure the sentence structure is correct - note: this is an extremely minor point.
2. The map in Figure 2 is a little hard to understand. First, the legend states that: “All populations are in Yosemite National Park (park boundary indicated by gray polygon)” – but I think the polygon is grey in the large map, but black in the inset. I would make the park boundary black in both maps. The legend also states: “In both maps, elevation is indicated by the colored hillshade layer (dark green = lowest elevation, white = highest elevation).” This is a little difficult for me to see too – there is dark green in the inset corresponding to the Central Valley, but in the larger map it looks like there is green on the top of the ridges, based on the perceived topography. Maybe I’m seeing the colors incorrectly.
3. Methods and associated supplemental materials
 - a. Line 348-354. This explanation is confusing because of the location of the ‘(primary period)’ reference. As written, it seems like the authors are saying that a single day is the primary period - only after you read the supplemental information do you realize that they are using a site visit as the primary period. I think it’s good that the authors used the primary/secondary period terminology that is common for robust design CMR surveys, but I think it would be better and more correct to say:

‘During 2006-2012, we conducted CMR surveys on a single day per site visit (primary period)... During 2013-2022, we generally used a robust design where all habitats were searched during several consecutive days (median number....) during each visit...’

In addition to the median number of secondary periods per primary period, it would also be nice to know the mean (or median) number of visits (primary periods) per summer active season during the study. At first, I thought that there was only 1 primary period per year, but the supplementary information clearly states: “Multiple primary periods can occur during a summer active season (typically July-August, but in some years as early as May and as late as September).” This would be a nice place to give the mean (or median) number of visits (primary periods) per summer active season.

b. Population viability model description.

- i. The authors have 7 state variables which they define as ‘density’ of individuals in different life history states. Is this really density (individuals per unit area) or just the number of individuals.
- ii. Additionally, are the authors modelling both males and females, or just females. Based on the model structure it looks like they are modelling females, as there is no adjustment for the proportion of females in the population. This should be clarified. If I am correct, then shouldn’t fecundity be expressed as the number of female eggs?
- iii. I am also a little confused at what they consider ‘recruitment’ in this model. I understand the survival probability, indicated by sigma notation (σ), but none of the parameters listed in Supplementary Table 1 are termed recruitment. However, in both the text the term is used in multiple places, e.g., lines 106, 102, 458 and figure legends (Fig 5), but I’m sure it means different things in these places. For this reason, I think the authors should clarify their use of the term, especially in the viability modelling. To me, recruitment is conditional on survival so I would tend to equate it with $(1 - pJ1)$ or the probability that given a juvenile survives it ‘recruits’ or transitions into an adult frog. I don’t think that is how the authors are using it in this model. I think they could refrain from using it at all in this section.

4. Given the page limitations employed by the journal, the authors must make decisions about what information to supply. Personally, I would have liked to see a little more information on the variation in recruitment (number of new adults a population) and population growth (change in abundance over time) among recipient sites. These are estimated (line 106; Supplementary Material) but not reported from their Bayesian Jolly-Seber CMR model. Instead, the authors chose to use limited demographic estimates from reintroduced populations (adult survival only) in a population viability model to explore variation in population viability over 50 years. The only thing that varies across their 12 reintroduced populations in this exercise is adult survival; the other parameters are the same across populations (Supplementary Table 1). In essence, this exercise shows how population viability varies with adult survival. It’s really important to choose reintroduction sites where adult annual survival exceeds 0.50, preferably 0.60, and the authors make a case that survival 1-year post-translocation is a good proxy for habitat quality; however, part of that case is assuming that all other demographic parameters are the same among populations, which is probably not true. The elasticity and sensitivity analyses associated with the viability analyses do suggest that adult survival is certainly the most important parameter (almost always true in vertebrate systems). Still, the decision of what to present given limited space is up to the authors and just because it does not match what this one reviewer would have preferred does not detract from the quality and quantity of work demonstrated in this paper. This is a really good paper and definitely worthy of publication in Nature Communications.

5. Lines 221-233. This is very true – congratulations!!

6. Line 252. I suggest revising this sentence; frog survival at natal (donor) sites was never reported. If you are referring to the difference between translocated and naturally recruited adults at the 3 recipient locations (Supplementary Fig. 3) I would clarify that.

Reviewer #3

(Remarks to the Author)

General comments:

This is a long-term dataset, in many ways exceptionally long for in depth amphibian studies, and a valuable manuscript on a priority topic for biodiversity conservation. Bd research has been criticised for missing direct conservation applications in the field, which makes this study particularly novel and insightful, despite focusing on a single species. The results are very encouraging and give much cause of cautious optimism for what has often appeared to be an almost intractable problem for amphibian conservation but while I understand the specificity of your results it would be good to have at least a paragraph that looks at the potential implications for other species and important steps forward that could be concluded from your own work.

In my opinion the text is clear and very well-written and I do not have much to add in relation to that. There is some repetition between introduction and discussion so please be aware of that. As a general comment, you mention that the three donor populations are large, with several hundred adults and that you have documented the genetic structure but I suggest it would be useful to briefly discuss the potential implications of the reintroductions from a genetic perspective and any potential monitoring follow-up of that aspect and mitigation strategy if required. There is a reasonable potential for creating a genetic bottleneck as a result of these reintroductions in the absence of some kind of genetic monitoring. Also, it would be useful to briefly explain why are reintroductions important (and cost-effective) instead of simply waiting for these recovering populations to naturally expand over time.

The methods look sound but I have limited experience in this type of multilevel Bayesian modelling.

Detailed comments

Title- Just a suggestion but for me the title would benefit from mentioning the species. Your work obviously has wider implications but remains currently very species-specific.

Line 155- for me this aspect of the apparent high variability linked to donor population is important and I suggest should be highlighted both in the results and especially in the discussion as a separate point. Would the reintroductions be feasible in terms of achieving long term population viability with the lowest survival based on donor population?

Line 266- I suggest more caution in this statement and perhaps combining with the following sentence, which seems a reasonable explanation but remains based on a relatively small number of sites and reintroduction events. I would strongly expect specific weather patterns to also play an important role in any given year, even if these might be rarer events that you have not yet captured.

Line 288- for me the value in this call for additional research warrants some brief but specific details about what that would entail. Otherwise, researchers naturally always call for additional research.

Lines 298-301- I think this is important and relevant in the wider amphibian conservation context, where juveniles at this crucial life stage are often the specifically missing dataset because they are harder to monitor. This is in many ways a failure of the sector and the opposite compared to say birds or mammals, yet has become inbuilt in much amphibian research and is worth highlighting as it has important consequences for progressing practical conservation.

Lines 310-315 I suggest you could be more forward looking here and suggest some critical steps forward for interpreting the success of your results for advancing the conservation prospects for other similarly threatened amphibian species. What are the crucial considerations that should inform their strategies and how would you go about it? It might also be useful to also briefly mention the relevance of your results for reintroductions from the perspective of captive breeding and amphibian arks given that the current approaches would not facilitate any kind of resistance building (if at all possible).

Line 317- why would it though? Now that you have developed a solid understanding of the process, could you not strategically reintroduce populations to facilitate ongoing natural colonisation once there are sufficient numbers to create some wider population resilience?

Line 337- this is an important methodological point and which requires a bit more substance in terms of how was absence determined, over what period, etc

Version 1:

Reviewer comments:

Reviewer #2

(Remarks to the Author)

I have reviewed the authors responses to my previous comments - all minor - and I am satisfied with their responses and changes to the manuscript. I have no further comments.

I will reiterate that I believe this is an extremely important paper and constitutes the largest reintroduction effort to date for an

amphibian challenged by Bd, both in terms of the number of reintroductions and their subsequent success. I highly recommend the paper for publication in Nature Communications.

Reviewer #3

(Remarks to the Author)

I had some trouble with the review of the revised manuscript version as there doesn't seem to be a track changes version, which would make the reviewing process simpler. That said, looking at the line by line revisions, I am satisfied that the authors have done a careful job and the manuscript has improved as a result in terms of clarity and overall conclusions. As before, this is a substantial piece of work and I look forward to seeing this published and used in future conservation work focused on amphibians.

Response to Reviewer Comments

MS # NCOMMS-24-29988

Our point-by-point response to all comments from the three reviewers is provided below. Reviewer comments are in black and our responses are in blue.

Reviewer 1

This is a brilliant manuscript that advances our understanding and ability to mitigate the impact of the worst disease to impact vertebrates, chytridiomycosis. The main findings are that reintroduced frogs sourced from populations of the Sierra Nevada yellow-legged frog (*Rana sierrae*) that are recovering from initial outbreaks and have increased resistance against chytridiomycosis are able to reestablish populations in the presence of the pathogen, Bd. The authors also investigated factors affecting reintroduction success and found that site, cohort, and frog attributes were important. They modelled the viability of reintroduced populations suggesting that many have a low probability of extinction over 50 years.

The work has major significance to the growing fields of chytrididomycosis, wildlife diseases, mitigation of threatening processes, conservation, reintroduction of species and related fields. The work is original and the first to demonstrate reintroduction of resistant frogs as a viable long term solution for recovering amphibian populations from the impacts of chytridiomycosis. The evidence presented supports the conclusions of the authors and the methodology is sound. The figures are excellent.

Some suggestions for improvement of the manuscript:

1. I appreciate the arguments made in favour of frog resistance rather than site characteristics being the reason for translocation success. Perhaps the strongest argument is that non resistant frogs were extirpated from these sites previously and would be expected to be extirpated again unless they had evolved resistance and this could be stated in the discussion.

We added a sentence to the Discussion to make the point highlighted by the reviewer more strongly (lines 236-240).

2. The manuscript could possibly be improved by recognising that although many species have failed to evolve resistance, this may occur over a longer time frame or alternatively, species could be assisted to evolve resistance through selective breeding programs etc (Kosch et al 2022).

We added additional text to the Discussion to make the point suggested by the reviewer (lines 347-351).

3. Similarly, the manuscript could perhaps expand on the barriers to natural recovery and spread of resistant populations and the importance of considering elevational disease gradients and connectivity when choosing reintroduction sites (Bell et al 2020).

The reviewer highlights several points in this comment. We added a paragraph to the Discussion that addresses several of these points (lines 316-334; see also our response to the final “general comment” made by Reviewer 3).

4. Perhaps the authors could make a point in the discussion regarding the importance of sustained effort and using adaptive management techniques. This approach ensured they improved their ability to choose recipient sites with higher habitat quality and ultimately led to improved translocation success.

We added additional sentences to the Discussion to make the point suggested by the reviewer (lines 345-347).

5. Finally, perhaps the conclusions could be a little more hopeful given the study’s results along with other current efforts at mitigating the threat of chytridiomycosis (Berger et al 2024, Waddle et al 2024).

We agree with the reviewer that important progress is being made to mitigate the impacts of chytridiomycosis on the world’s amphibians. Nonetheless, many challenges remain. In writing the final paragraphs of the Discussion, we therefore didn’t want to oversell the significance of our study and that of other recent studies (as cited by the reviewer). However, we agree that the paragraph in question could provide an outlook that is a bit more hopeful. We made several changes to this paragraph to accomplish that (lines 335-355).

Congratulations to the authors on a very impressive body of work and most importantly, wonderful results for amphibian conservation. Extremely well done!

We appreciate the reviewer’s enthusiastic endorsement of our study.

Reviewer 2

This is an extremely important paper – the authors have conducted extraordinary conservation work on an amphibian species known to be highly susceptible to the lethal fungal pathogen (*Batrachochytrium dendrobatidis*; “Bd”). Their efforts to reintroduce the Sierra Nevada yellow-legged frog (*Rana sierrae*) to locations where it had formally been extirpated using adults from naturally recovering populations is extremely promising. To my knowledge, this paper constitutes the largest reintroduction effort to date for an amphibian challenged by Bd, both in terms of the number of reintroductions and their subsequent success. I have few minor comments (below) and highly recommend publishing this paper. It will certainly be highly cited and serve as a model for other amphibian systems impacted by the pathogen. Additionally, I think the methods and associated data analyses are well explained and appropriate (see minor comments/suggestions below).

Minor comments/suggestions: (these are in no order)

1. Line 90-98. It seems like the sentences that discuss the three primary objectives may not be complete sentences? It seems like they might need another word: e.g., First, ‘we’ determined whether...Second, ‘we’ identified.... Etc. Maybe authors mean to provide a list. We had three primary objectives (1) determine whether...; (2) identify important site,... (3) estimate the probability of

persistence...but that is a lot to include in a single sentence, so I understand (and like) breaking it apart, but I am just not sure the sentence structure is correct - note: this is an extremely minor point.

The sentences in question (lines 90-98) are complete sentences, and the reviewer's suggestion to add "we" to each sentence changes the sentences from objectives to descriptions of the three components of the study. We did not make any changes in response to this comment, but we are open to any suggestions to improve the structure of these sentences.

2. The map in Figure 2 is a little hard to understand. First, the legend states that: "All populations are in Yosemite National Park (park boundary indicated by gray polygon)" – but I think the polygon is grey in the large map, but black in the inset. I would make the park boundary black in both maps. The legend also states: "In both maps, elevation is indicated by the colored hillshade layer (dark green = lowest elevation, white = highest elevation)." This is a little difficult for me to see too – there is dark green in the inset corresponding the Central Valley, but in the larger map it looks like there is green on the top of the ridges, based on the perceived topography. Maybe I'm seeing the colors incorrectly.

Thank you for pointing out these difficulties in interpreting the map. In the revised version of Figure 2, the polygon indicating the park boundary is now displayed in black in both the main map and the inset map. In addition, we added clarifying text to the figure legend to describe the colors and their meaning.

3. Methods and associated supplemental materials

- a. Line 348-354. This explanation is confusing because of the location of the '(primary period)' reference. As written, it seems like the authors are saying that a single day is the primary period - only after you read the supplemental information do you realize that they are using a site visit as the primary period. I think it's good that the authors used the primary/secondary period terminology that is common for robust design CMR surveys, but I think it would be better and more correct to say:

'During 2006-2012, we conducted CMR surveys on a single day per site visit (primary period).... During 2013-2022, we generally used a robust design where all habitats were searched during several consecutive days (median number....) during each visit...'

We updated this sentence as the reviewer suggested (lines 390-392).

In addition to the median number of secondary periods per primary period, it would also be nice to know the mean (or median) number of visits (primary periods) per summer active season during the study. At first, I thought that there was only 1 primary period per year, but the supplementary information clearly states: "Multiple primary periods can occur during a summer active season (typically July-August, but in some years as early as May and as late as September)." This would be a nice place to give the mean (or median) number of visits (primary periods) per summer active season.

In the reviewed version of the manuscript, we actually included the median, minimum, and maximum number of visits to each translocated population per summer active season (lines 345-346). However, it may not have been clear that a "visit" is equivalent to a "primary period". Therefore, we added some additional text to the sentence in question to clarify this point (revised manuscript: lines 385-388).

b. Population viability model description.

- The authors have 7 state variables which they define as 'density' of individuals in different life history states. Is this really density (individuals per unit area) or just the number of individuals.

Good point. We are modeling abundance per population so we have replaced the word density with abundance to avoid confusion over the units of area (lines 468-472).

- Additionally, are the authors modelling both males and females, or just females. Based on the model structure it looks like they are modelling females, as there is no adjustment for the proportion of females in the population. This should be clarified. If I am correct, then shouldn't fecundity be expressed as the number of female eggs?

Thank you for the careful reading of our model. We are modeling the entire population, not just females, and are assuming a 50:50 sex ratio. This was not explicit in our equations and we have thus updated our viability model equation to explicitly include a parameter for sex ratio (ρ ; line 477). We also added a description of this parameter in Supplementary Table 1. While addressing this comment, we noticed that we were not consistent with modeling females vs. total population between the deterministic and stochastic models. We have updated the deterministic model to be exactly analogous to the stochastic model (i.e., both are now modeling total population). Our conclusions regarding viability are unchanged.

- I am also a little confused at what they consider 'recruitment' in this model. I understand the survival probability, indicated by sigma notation (σ), but none of the parameters listed in Supplementary Table 1 are termed recruitment. However, in both the text the term is used in multiple places, e.g., lines 106, 102, 458 and figure legends (Fig 5), but I'm sure it means different thing in these places. For this reason, I think the authors should clarify their use of the term, especially in the viability modelling. To me, recruitment is conditional on survival so I would tend to equate it with $(1 - p)J_1$ or the probability that given a juvenile survives it 'recruits' or transitions into an adult frog. I don't think that is how the adults are using it in this model. I think they could refrain from using it at all in this section.

Good question. This was a point of some discussion within our group. For a juvenile to recruit to an adult, it has to survive, and thus σ_{J_1} encompasses survival and recruitment. This is why we used the language of "survival and recruitment" to describe this parameter in the text. However, we see how this could be confusing. Because we do not lose much biological interpretation by just referring to σ_{J_1} as "survival" we have made this change throughout the text as suggested by the reviewer.

- Given the page limitations employed by the journal, the authors must make decisions about what information to supply. Personally, I would have liked to see a little more information on the variation in recruitment (number of new adults a population) and population growth (change in abundance over time) among recipient sites. These are estimated (line 106; Supplementary Material) but not reported from their Bayesian Jolly-Seber CMR model. Instead, the authors chose to use limited demographic estimates from reintroduced populations (adult survival only) in a population viability model to explore variation in population viability over 50 years. The only

thing that varies across their 12 reintroduced populations in this exercise is adult survival; the other parameters are the same across populations (Supplementary Table 1). In essence, this exercise shows how population viability varies with adult survival. It's really important to choose reintroduction sites where adult annual survival exceeds 0.50, preferably 0.60, and the authors make a case that survival 1-year post-translocation is a good proxy for habitat quality; however, part of that case is assuming that all other demographic parameters are the same among populations, which is probably not true. The elasticity and sensitivity analyses associated with the viability analyses do suggest that adult survival is certainly the most important parameter (almost always true in vertebrate systems). Still, the decision of what to present given limited space is up to the authors and just because it does not match what this one reviewer would have preferred does not detract from the quality and quantity of work demonstrated in this paper. This is a really good paper and definitely worthy of publication in Nature Communications.

We appreciate the reviewer's thoughtful point. It is worth noting that the viability analysis also explores variation in σ_{J_1} , a parameter that our sensitivity analyses indicate is likely the second-most important parameter for viability following adult survival. So we are actually exploring viability across environmental conditions that affect the two most important parameters for population growth. While we agree that other vital rates likely vary to some degree across populations, we have good empirical and modeling evidence that the two parameters we explore in detail are the most important for long-term viability. The reviewer did not request any changes in response to this comment, and we did not make any.

- Lines 221-233. This is very true – congratulations!!

We appreciate the reviewer's acknowledgement and agreement with this important point of ours.

- Line 252. I suggest revising this sentence; frog survival at natal (donor) sites was never reported. If you are referring to the difference between translocated and naturally recruited adults at the 3 recipient locations (Supplementary Fig. 3) I would clarify that.

As the reviewer suggested, we added a reference to Supplementary Fig. 3 in this sentence (lines 256-258).

Reviewer 3

General comments:

This is a long-term dataset, in many ways exceptionally long for in depth amphibian studies, and a valuable manuscript on a priority topic for biodiversity conservation. Bd research has been criticised for missing direct conservation applications in the field, which makes this study particularly novel and insightful, despite focusing on a single species.

The results are very encouraging and give much cause of cautious optimism for what has often appeared to be an almost intractable problem for amphibian conservation but while I understand the specificity of your results it would be good to have at least a paragraph that looks at the potential implications for other species and important steps forward that could be concluded from your own work

We added the suggested text to one of the final paragraphs of the Discussion (lines 335-355). See also our response to Reviewer 1 - Comment 4 above.

In my opinion the text is clear and very well-written and I do not have much to add in relation to that. There is some repetition between introduction and discussion so please be aware of that.

We edited the Discussion to minimize any repetition between the Introduction and Discussion. Any remaining repetition reflects our interest in reinforcing points in the Discussion that were made initially in the Introduction. When specific points are repeated in both sections, we have ensured that the text differs between the two sections of the manuscript.

As a general comment, you mention that the three donor populations are large, with several hundred adults and that you have documented the genetic structure but I suggest it would be useful to briefly discuss the potential implications of the reintroductions from a genetic perspective and any potential monitoring follow-up of that aspect and mitigation strategy if required. There is a reasonable potential for creating a genetic bottleneck as a results of these reintroductions in the absence of some kind of genetic monitoring.

We appreciate this important point. We added text to one of the last paragraphs of the Discussion to address this issue (lines 351-355).

Also, it would be useful to briefly explain why are reintroductions important (and cost-effective) instead of simply waiting for these recovering populations to naturally expand over time.

This is another very important issue, and we thank the reviewer for highlighting it. We added a new paragraph to the Discussion that summarizes the inadequacy of natural recovery and the importance of translocations in allowing reestablishment of populations at the landscape scale (lines 316-334).

The methods look sound but I have limited experience in this type of multilevel Bayesian modelling.

Detailed comments

Title- Just a suggestion but for me the title would benefit from mentioning the species. Your work obviously has wider implications but remains currently very species-specific.

The title is currently at the recommended limit of 15 words. As such, although adding "*Rana sierrae*" to the title might be useful in some contexts, we opted to leave the title unchanged.

Line 155- for me this aspect of the apparent high variability linked to donor population is important and I suggest should be highlighted both in the results and especially in the discussion as a separate point. Would the reintroductions be feasible in terms of achieving long term population viability with the lowest survival based on donor population?

This is an important question, but we cannot address it directly using results from the current study. Our population viability analysis shows that long-term viability is significantly more likely when 1-year adult survival probability is >0.5 . In addition, there is strong support from the model that the lowest 1-year survival from donor population 72006 would not be sufficient for long-term viability (adult

1-year survival from donor population 72996 would not be sufficient for long-term viability (adult survival probability < 0.5) and that the lowest 1-year adult survival from 70459 and 70567 would be sufficient for long-term viability (adult survival probability > 0.5). Although it is tempting to conclude that donor populations 70459 and 70567 should be used in future translocations, our study design cannot disentangle the effects of recipient site and donor population on 1-year adult survival because these factors were not crossed (an inevitable logistical constraint when working on real landscapes and taking into account *R. sierrae* landscape genetics). We are currently conducting translocations in which frogs from several donor populations are added to a single site, and this design will allow us to separate these effects. Given our inability in the current study to estimate the independent effect of donor population on adult survival, we opted not to emphasize this issue, and instead will provide a detailed treatment of it in a subsequent paper that describes the results of the multi-donor translocations. However, we added a brief description of these multi-donor translocations to the Discussion (lines 291-297).

Line 266- I suggest more caution in this statement and perhaps combining with the following sentence, which seems a reasonable explanation but remains based on a relatively small number of sites and reintroduction events. I would strongly expect specific weather patterns to also play an important role in any given year, even if these might be rarer events that you have not yet captured.

We softened our assertion that 1-year frog survival is an effective proxy of longer-term survival and population viability by replacing the word "indicate" with "suggest" (lines 269-271).

Line 288- for me the value in this call for additional research warrants some brief but specific details about what that would entail. Otherwise, researchers naturally always call for additional research.

We replaced the text in question with a specific description of the study we are currently conducting to quantify the effects of donor population (lines 291-297; see also our response to a comment by Reviewer 3 above).

Lines 298-301- I think this is important and relevant in the wider amphibian conservation context, where juveniles at this crucial life stage are often the specifically missing dataset because they are harder to monitor. This is in many ways a failure of the sector and the opposite compared to say birds or mammals, yet has become inbuilt in much amphibian research and is worth highlighting as it has important consequences for progressing practical conservation.

We added a sentence to the end of this paragraph that emphasizes the critical need for better monitoring of this vulnerable life stage (lines 312-315).

Lines 310-315 I suggest you could be more forward looking here and suggest some critical steps forward for interpreting the success of your results for advancing the conservation prospects for other similarly threatened amphibian species. What are the crucial considerations that should inform their strategies and how would you go about it? It might also be useful to also briefly mention the relevance of your results for reintroductions from the perspective of captive breeding and amphibian arks given that the current approaches would not facilitate any kind of resistance building (if at all possible).

As described in our response to Reviewer 1 - Comment 4, we revised this paragraph to include specific

recommendations for ongoing and future efforts to recover Bd-impacted amphibians. The added text addresses some of the specific points made by Reviewer 3, including mention of methods such as captive breeding and genetic inventions to facilitate the development of frog resistance against Bd infection (lines 335-355).

Line 317- why would it though? Now that you have developed a solid understanding of the process, could you not strategically reintroduce populations to facilitate ongoing natural colonisation once there are sufficient numbers to create some wider population resilience?

We added a paragraph to the Discussion to elaborate on the limitations of natural recovery and the need for active management via reintroductions to reestablish frogs at the landscape scale (lines 316-334). The specific scenario that the reviewer mentions is described in some detail in that paragraph (see also our response to the “general comment” from Reviewer 3 above regarding natural recovery versus recovery via reintroductions).

Line 337- this is an important methodological point and which requires a bit more substance in terms of how was absence determined, over what period, etc

We added a sentence to this paragraph that briefly describes the methods we used to determine *R. sierrae* absence (lines 377-379).